# HOW DO DEEP CONVOLUTIONAL NEURAL NETWORKS LEARN FROM RAW AUDIO WAVEFORMS?

## ABSTRACT

Prior work on speech and audio processing has demonstrated the ability to obtain excellent performance when learning directly from raw audio waveforms using convolutional neural networks (CNNs). However, the exact inner workings of a CNN remain unclear, which hinders further developments and improvements into this direction. In this paper, we theoretically analyze and explain how deep CNNs learn from raw audio waveforms and identify potential limitations of existing network structures. Based on this analysis, we further propose a new network architecture (called SimpleNet), which offers a very simple but concise structure and high model interpretability.

## 1 INTRODUCTION

In the field of speech and audio processing, due to the lack of tools to directly process high dimensional data, conventional audio and speech analysis systems are typically built using a pipeline structure, where the first step is to extract various low dimensional hand-crafted acoustic features (e.g., MFCC, pitch, and formant frequencies) (Eyben et al., 2010). Although hand-crafted acoustic features are typically well designed, is still not possible to retain all useful information due to the human knowledge bias and the high compression ratio. To overcome these limitations, several prior efforts began to abandon handcrafted features and instead feed raw magnitude spectrogram features directly into the deep convolutional neural networks (CNNs) or deep recurrent neural networks (DNNs) (Hannun et al., 2014; Zheng et al., 2015; Ghosh et al., 2016; Badshah et al., 2017). These approaches yield significant performance improvements compared to hand-crafted feature-based approaches. Furthermore, a more recent trend in audio and speech processing is **learning directly from raw waveforms** (i.e., the raw waveform is fed directly into the deep CNNs), which provides a more thorough end-to-end process by completely abandoning the feature extraction step. These approaches have been shown to deliver superior performance compared to approaches using hand-crafted features (Sainath et al., 2015; Aytar et al., 2016; Trigeorgis et al., 2016; Thickstun et al., 2016; Dai et al., 2017).

Despite the encouraging progress into this direction, one major concern has not yet been addressed, which is the lack of understanding of how deep CNNs actually learn from an audio waveform. Although the mechanisms of how deep CNNs operate in computer vision applications have been quite clearly described in prior work (Zeiler & Fergus, 2014), this analysis cannot directly explain the internal operation and behavior of raw waveform processing, because images and audio files are quite different in their representation. For example, in computer vision, the first layer of CNNs usually extracts low-level features such as edges, while the following layers of CNNs usually capture high-level features such as shapes. However, there are no corresponding concepts of "edges" and "shapes" for audio waveforms. Therefore, it remains unknown what actual features CNNs learn from waveforms.

From a scientific standpoint, using deep CNNs as a black box for audio and speech processing is deeply unsatisfactory. Without a clear understanding of how and why CNNs can learn from raw audio waveforms, the development of better models, i.e., the design of the network architecture and the determination of hyper-parameters, is reduced to trial-and-error. In fact, the network architecture and hyper-parameter settings of existing approaches are similar to those of computer vision models. However, the architectures and settings that work well for computer vision tasks are not necessarily good choices for audio and speech tasks.

Due to the above-mentioned reasons, in order to further explore end-to-end audio and speech processing techniques, it is imperative to perform a thorough theoretical analysis and investigation of the inner workings of CNNs. As a first effort aiming to understand the inner workings of CNN models that learn directly from the raw waveform, our contributions are as follows. We theoretically analyze and explain how deep CNNs learn from raw audio waveforms from the perspective of machine learning and audio signal processing. Based on such analysis, we then 1) discuss the potential limitations of existing network structures, 2) unify the approaches based on spectrogram features and the raw waveform by showing that the former is actually a special case of the latter, and 3) propose a new end-to-end audio waveform learning scheme, which includes a set of components that are specifically designed for waveform processing. This new scheme features an extremely simple network architecture, but better performance compared to previous networks.

## 2 UNDERSTANDING HOW DEEP CONVOLUTIONAL NEURAL NETWORKS LEARN FROM RAW AUDIO WAVEFORMS

### 2.1 BASICS

Before discussing the details of our approach, we first briefly review some important concepts in audio signal processing that will be used in the remainder of this paper.

#### 2.1.1 FREQUENCY DOMAIN ANALYSIS OF AUDIO SIGNALS

Audio signals are time-variant and non-stationary signals. However, using short-time Fourier transform (STFT), for each short-time window, an audio signal can be decomposed into a series of frequency components. This allows us to analyze audio signals with respect to the frequency (i.e., frequency domain analysis) in addition to the time domain analysis.

#### 2.1.2 NYQUIST SAMPLING THEOREM AND EFFECTS OF ALIASING

The Nyquist sampling theorem can be stated as follows: the *sampling frequency should be at least twice the highest frequency contained in the signal*. This is expressed mathematically as:

$$f_s \geq 2 \times f_c,$$

where $f_s$ is the sampling rate and $f_c$ is the highest frequency contained in the signal. Specifically in audio processing, the sampling rate is usually selected following the Nyquist sampling theorem, e.g., the highest frequency of normal human speech is around 8kHz, thus a sampling rate of 16kHz is enough; the highest frequency in music is typically much higher and therefore, a sampling rate of 44.1kHz (or even higher) is frequently used.

Violating the Nyquist sampling (i.e., the sampling rate is smaller than twice the highest frequency in the signal) will lead to the aliasing effect. In the frequency domain, the signal components whose frequency is over the limit will be cut off, mistakenly sampled as low-frequency signal components, and then mixed with signal components of low frequency. In the time domain, this is then expressed as a distortion.

#### 2.1.3 CONVOLUTION THEOREM

The convolution theorem can be stated as follows:

$$f * g = \mathcal{F}^{-1}\{\mathcal{F}\{f\} \cdot \mathcal{F}\{g\}\},$$

where $f$ is the filter, $g$ is the input signal, $*$ is the convolution operation, $\cdot$ is the point-wise multiplication operation, and $\mathcal{F}$ and $\mathcal{F}^{-1}$ are the Fourier and inverse Fourier transformations, respectively. According to the convolution theorem, conducting a convolution operation on the input signal and the filter in the time domain is equivalent to multiplying their Fourier transformations point-wise in the frequency domain.

## 2.2 UNDERSTANDING CNNS

In this section, we begin our discussion on the inner workings of the end-to-end waveform processing model by analyzing the functions of commonly used network components one by one. Specifically, we center our discussion around two representative techniques: SoundNet (the 8 layer version) (Aytar et al., 2016) and the work presented in (Trigeorgis et al., 2016), which we will refer to as WaveRNN (note that WaveRNN uses a stack of CNNs as front-end layers and one or more RNNs as final layers). The architectures of SoundNet and WaveRNN are shown on the left in Figure 5.

### 2.2.1 PREPROCESSING

Several prior techniques performed a *windowing step* before feeding the waveform into the network, e.g., WaveRNN first slices the audio into 40ms chunks, inputs each chunk to a stack of convolutional and pooling layers, and then passes the output of each window into the recurrent layers, while maintaining the correct temporal sequence. An advantage of this step is that it explicitly controls the time resolution of the feature, i.e., a smaller window will increase the time resolution and vice versa. This is particularly useful when we want to consider task-specific requirements, e.g., in the *speaker emotion recognition* task, we need high temporal resolution, because short-term details are important for inference. In contrast, in the *speaker gender recognition* task, high time resolution is not required, because the inference relies less on the temporal details. However, the windowing step is not mandatory (e.g., SoundNet does not have a windowing step), because the time resolution of the features can be implicitly controlled by the number of temporal pooling layers and the pooling size. In either case, the waveform will then be fed into stacked convolutional layers.

### 2.2.2 STACKED CONVOLUTIONAL LAYERS

Stacked convolutional layers are one of the most commonly used architectures for computer vision tasks and they also appear in almost all state-of-the-art end-to-end waveform processing models. We are interested in understanding which features stacked convolutional layers can learn from the audio waveform. For ease of discussion, we begin with a simple network, which contains just two convolutional layers, where both have two filters (each filter of the second layer has two subfilters for each input channel). The filters are simple highpass, lowpass, and bandpass filters, linear activation is used in the network, and no pooling layers have been added.

We then feed an audio waveform of 6 seconds (with a sampling rate of 16kHz) into the network. Figures 1 and 2 show the data flow in the time domain and the frequency domain, respectively. Note that according to the convolutional theorem, the convolutional operation in the time domain is equivalent to a filtering operation in the frequency domain. We find that it is more informative in the frequency domain, where we can observe that the original audio is first filtered into 0-4kHz and 4-8kHz regions at the first layer, and then goes through another round of filtering and adding operations in the second layer. Finally, Output (2,1) contains the frequency components of 2-4kHz and 6-8kHz of the input audio waveform and Output (2,2) contains almost no frequency components (and therefore also has very little energy in the time domain). The outputs of the subsequent layers are also combinations of signal components in some frequency band since they are generated through some further filtering and adding operations based on the output of the second layer.

In other words, the output of stacked convolutional layers are a set of linear combinations of frequency components of the original audio signal and the coefficients of the combinations are determined by the characteristics of the filters, which are learned by the optimization algorithm. This point is simple, yet particularly interesting, because it shows that the stacked convolutional layers are learning what the informative and discriminative spectra are with regards to the task, by optimizing the convolutional filter. Historically in audio processing, researchers performed a lot of work manually to find the informative frequency spectra and corresponding filters. One famous example is the Mel-scale filter bank and its corresponding feature Mel-frequency cepstral coefficients (MFCCs) (Davis & Mermelstein, 1980). But the most informative spectra and spectra combinations are actually task-specific and hard to be modeled manually. And further, this characteristic of waveform-based approaches is very different from spectrogram-based approach (Zheng et al., 2015), i.e., the spectrogram-based approach uses a set of fixed filters to filter out spectra from the signal and then feed the magnitude or energy of them into the neural networks, while the filters of waveform-based approach are learned and optimized by the neural network itself. This might

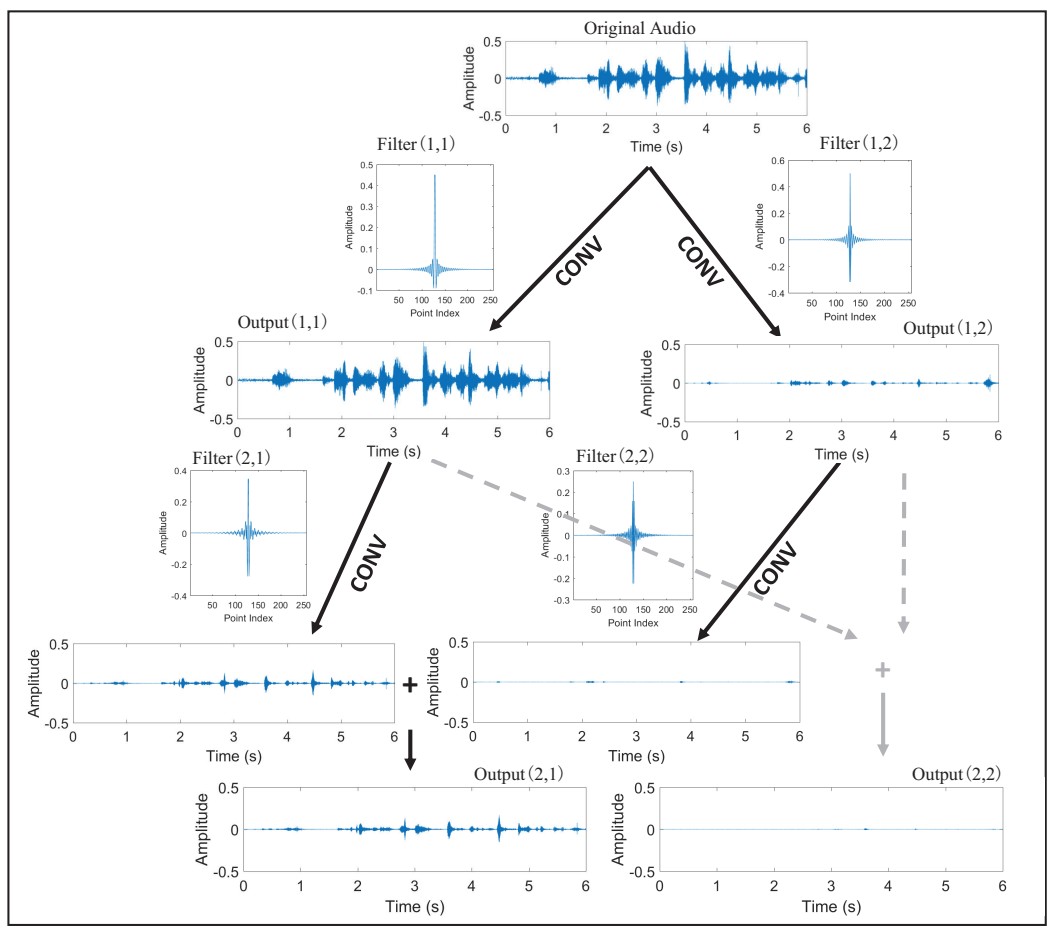

Figure 1: The data flow in the frequency domain. The generation progress of Output (2,2) has been omitted.

be the reason leads to the performance difference between the waveform-based approach and the spectrogram-based approach. We empirically demonstrate this in Section 4.

With this finding, the question arises why we need stacking convolutional layers for waveform processing models if the same output can be generated by a single convolutional layer, e.g., Output (2,1) can be generated using a single multi-band bandpass filter from the original audio and Output (2,2) can be generated using a single all stop filter. From the perspective of the deep neural network, the large number of layers actually increases the difficulty of the optimization, e.g., potentially leading to the vanishing gradients problem. We also discuss this question later in the paper.

### 2.2.3 POOLING LAYERS

Another network component that is used in almost all CNNs is the pooling layer. In SoundNet and WaveRNN, the pooling layers are also heavily used. WaveRNN has one pooling layer following each convolutional layer (for a total of 8 pooling layers). SoundNet contains 3 pooling layers, but the pooling size is larger (2 layers with pooling size of 8 and 1 layer with pooling size of 4). One intuition from the computer vision field is that pooling layers, when working together with convolutional layers, can hierarchically extract different levels of features such as edges, shapes, and objects. One would therefore expect that waveform processing is similarly able to extract different levels of features. However, pooling layers actually help very little with the processing of the raw waveform and may even hurt the performance if they are not properly applied.

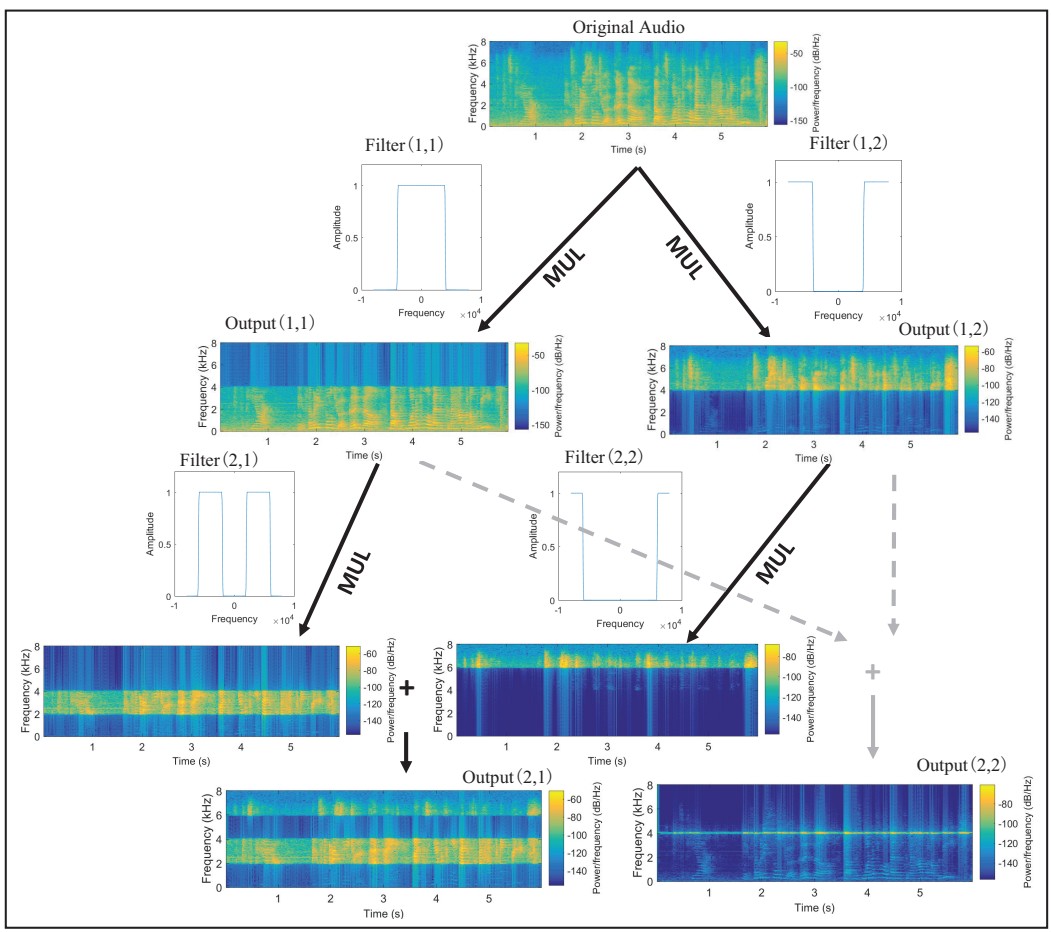

Figure 2: The data flow in the frequency domain. The convolution operation in the time domain is equivalent to the point-wise multiplication (filtering) in the frequency domain. The generation progress of Output (2,2) has been omitted.

Theoretically, according to the Nyquist sampling theorem, performing temporal pooling in audio processing (i.e., pooling across time) is a downsampling operation, which will further lead to the **aliasing effect**, where the frequency components that are more than half of the new sampling rate will be mistakenly sampled as low-frequency components and mixed into the real low-frequency components. Therefore, the convolutional layers after the pooling layer cannot perform filtering effectively since the aliasing high-frequency components and the real low-frequency components are completely indistinguishable. Further, the aliasing effect grows exponentially with the number of pooling layers. One might argue that the aliasing effect also appears in image processing and the CNNs may have the capability to learn even when aliasing does occur. However, this is not the case, because in computer vision, the spatial frequency is not the only information the model can use and aliasing is less likely to happen since the spatial frequency of images is usually low. In contrast, as stated in Section 2.2.2, frequency information is essential for audio models. Further, the sampling rate of audio is usually just above twice the highest frequency, which means that only a few pooling layers will lead to the aliasing effect. The aliasing effect is non-reversible, thus there is no way that the convolutional filters can distinguish real frequency components from aliased frequency components and perform effective filtering.

An example of the aliasing effect is shown in Figure 3, where the network is identical to that of Figures 1 and 2 with the exception of one max pooling layer that is attached to each convolutional layer. Output (1,2) is generated by a highpass filter on the input audio that originally should contain only few frequency components between 0-4kHz (as shown in Figure 2), but after the pooling oper-

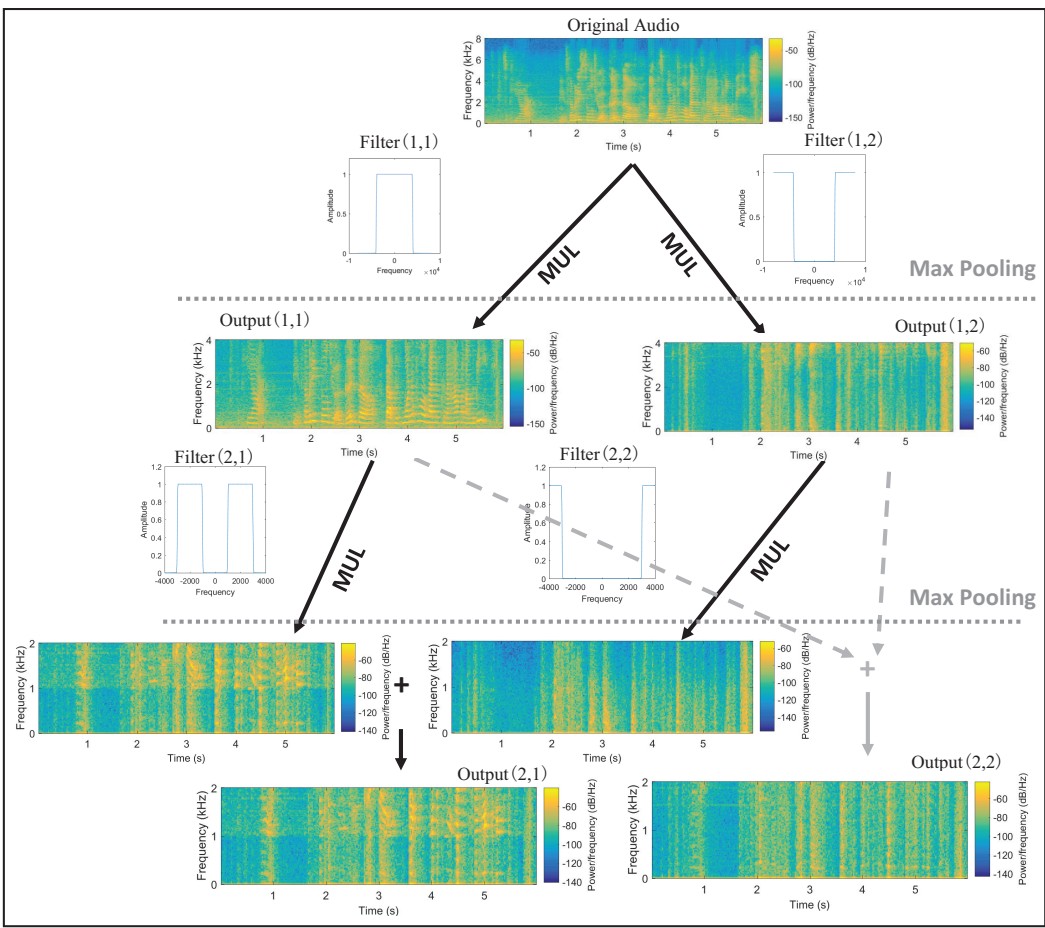

Figure 3: An example of the aliasing effect caused by the pooling layer. Note the change of the range on the frequency (y) axis of the spectrogram: higher frequencies of both the filter and the signal are cut off according to the reduction of the sampling rate.

ation, the high-frequency components are mistakenly sampled into 0-4kHz and further mixed with real 0-4kHz components in the following convolution operation of the second layer. That is, starting with the second layer, 0-4kHz components and 4-8kHz components of the input audio become completely indistinguishable in the model, and the following convolutional layers will further lead to an accumulation of this error.

Based on this analysis, the pooling layers will lower the effectiveness of the following convolutional layers. As a consequence, one possible way to use them is by placing them after the last convolutional layer. In this case, they work purely as a compressor. As mentioned in Section 2.2.2, the output of the convolutional layers is a set of signal components. Performing max or average pooling on them is an approximation of the energy of each component over time.

### 2.2.4 HIGH-LEVEL LAYERS

The output of the convolutional layers is usually still very high-dimensional (e.g., the output of the last convolutional layers of both WaveRNN and SoundNet have thousands of dimensions). Thus, the output of a convolutional layer can be further processed by RNNs (such as in WaveRNN) or other classifiers (such as in SoundNet). Actually, the spectrogram-based approach is very similar to the waveform-based approach beginning with higher-level layers. The difference is mainly in the front-end layers.

### 2.2.5 NON-LINEAR COMPONENTS

In practical networks, the activation function is usually not a simple linear activation as used in our example. One common choice is the Rectified linear units (ReLU) (Nair & Hinton, 2010). ReLU is equivalent to a half-wave rectifier in audio signal processing, which will add harmonic frequency components to the output, but its effect is small compared to the aliasing effect. There are also non-linear activation functions such as the sigmoid function. The sigmoid function suppresses large values in the time domain, which is equivalent to suppressing the high energy frequency components according to the Parseval theorem. ReLU and non-linear activations can improve the network performance, but they are not the main factors in the inner workings of CNNs.

### 2.2.6 SUMMARY

In this section, we discussed that the output of convolutional layers is actually a set of linear combinations of frequency components of the original audio signal, while the coefficients of the combinations are determined by the characteristics of the filters, which in turn are optimized by the network itself. The pooling layers can lower the dimension, but will also lead to the aliasing effect, which lowers the effectiveness of the following convolutional layers. A pooling layer attached to the last convolutional layer can be considered as an approximation of the energy of the signal components output by the convolutional layer over time. Thus, the connection and difference between the spectrogram-based approach and the waveform-based approach is as follows: both of them feed the energy or magnitude of a set of signal components of specific spectra to high-level layers as features, but the spectra of the signal components are fixed by using designed filters (e.g., Mel-scale filter bank) in the spectrogram-based approach as a preprocessing step, while in the waveform-based approach, the spectra of the signal components are learned internally in the neural network. In this sense, the spectrogram-based approach is actually a special case of the waveform-based approach, i.e., if we set the filters in the waveform approach fixed and untrainable, then the waveform approach will downgrade to the spectrogram approach.

Based on our analysis, we are skeptical about the effectiveness of stacking the convolutional layers and pooling layers in audio processing, because the output of multiple convolutional layers actually can also be obtained by a single convolutional layer and the aliasing effect caused by the pooling layer will further impact the performance. These two concerns motivate our proposed SimpleNet architecture, which will be introduced in the following section.

## 3 SIMPLENET

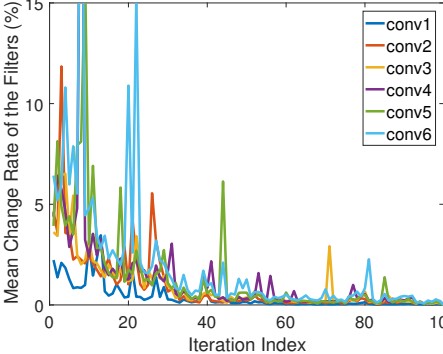 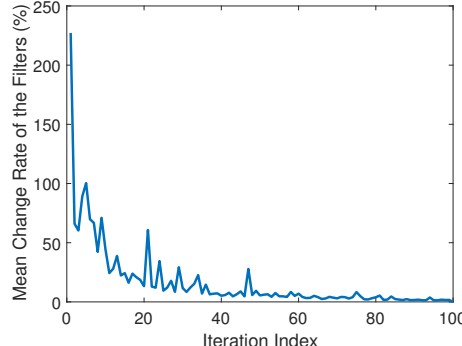

Figure 4: Comparison of the mean change rate of the convolutional filters (i.e., weights) of SoundNet and SimpleNet during training in an identical speaker emotion recognition test. Left: the mean change rate of the first 6 convolutional layers of SoundNet. Right: the mean change rate of the convolutional layer of SimpleNet. The mean change rate is defined as the mean absolute value of the change rate of the value of each weight compared to the last iteration, which is defined as $(value_i - value_{i-1})/value_{i-1}$ for the $i$th iteration. The changing rate reflects if the convolutional filters are effectively trained. Filters of SimpleNet are trained more effectively.

As shown in Figure 4 (left), in our experiments, we observe that the change of the convolutional filters of SoundNet during training is small (e.g., the value of the filters vary little from their initial states, which are random values), which means that these convolutional layers are not trained effectively, even though batch normalization (Ioffe & Szegedy, 2015) is used between layers. One possible reason is that the aliasing effect makes it difficult for the filters to extract useful features (and hence help improve the performance), no matter what value they use, i.e., the gradients of the loss function with regard to these weights are small in the optimization. The large number of layers of SoundNet might also increase the difficulty of the training. When the front-end convolutional layers cannot extract discriminative features from the raw waveform, the model then heavily relies on the fully-connected layer, which will cause more overfitting. In fact, we do observe more severe overfitting for SoundNet in our following experiments.

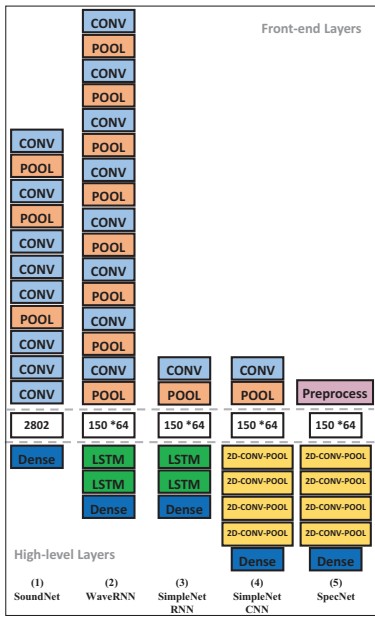

Figure 5: Comparison of the various architectures discussed in this paper. The output dimension of the front-end layers (when the network input is a 6-second audio with the sampling rate of 16kHz) is shown between the dashed lines for each network.

In order to avoid these shortcomings, we propose a new network architecture called **SimpleNet**. SimpleNet features an extremely simple structure with only one convolutional layer and one pooling layer as the front-end layers compared to more than 10 layers in SoundNet and WaveRNN. This design avoids the aliasing effect problem by placing the pooling layer after the only convolutional layer, thus no further filtering is performed and the network will therefore not be affected by the aliasing effect. A single convolutional layer (instead of a stack of convolutional layers) is used, which greatly reduces the number of layers and is expected to make optimization easier. This design is based on our observation in Section 2.2.2 that the same output of multiple convolutional layers can be achieved by a single one. Note that SimpleNet has a windowing step that is identical to WaveRNN.

We further propose a new convolutional filter initializer specifically designed for waveform processing. That is, we manually design a set of digital filters with the desired characteristics and use them as the initial states of the convolutional filters. One advantage of this approach is that we can perform task-specific initialization of the network, e.g., for speech tasks, usually the lower frequency components are more informative and therefore we can initialize the convolutional filters with a set of bandpass filters with low pass bands. In contrast, for music tasks, we can initialize the filters according to the music range. We can also initialize the convolutional filters with the Mel-scale frequency bank. This manual initialization does not have to be accurate, because the neural network will further learn the filters using the processed data. The proposed initializer works like a regularizer of human domain knowledge in the training phase while implementing a real regularizer controlling

the behavior of filters in the frequency domain is hard. As described in Section 2.2.6, the difference between spectrogram-based approach and waveform-based approach is that the former uses fixed filters designed manually, while the latter uses filters learned by the neural network. In this sense, the proposed initializer helps us make good use of the strengths of the different approaches by integrating domain knowledge and learning capability.

Due to its simple structure, SimpleNet has a limited number of hyper-parameters, which are also easy to tune. The number of convolutional filters can be tuned according to the number of features needed: more complex tasks typically require more filters (a typical value of filters used in our work is 64). The size of the convolutional filter determines the frequency resolution of the filters (i.e., how accurate the filtering can be performed); higher values should be used if the task is frequency-sensitive, e.g., we use a value of 256, which is equivalent to a frequency resolution of 31.25Hz when the sampling rate is 16kHz.

The pooling layer has a pooling size equal to the size of the convolutional layer output and we use average absolute pooling (i.e., calculating the mean of the absolute value of each point of the entire input). Its purpose is to simply estimate the energy of each of the convolutional layer outputs, which is a set of signal components of specific spectra. It can be replaced by a mean squared pooling layer that calculates the energy directly, but note that the square function might make the output data become too large or too small for the next layer.

SimpleNet can work with any high-level layers, e.g., we can implement SimpleNet-RNN by connecting SimpleNet with two LSTM layers (which is identical to WaveRNN). A slightly more complex implementation is to explicitly shape the output of SimpleNet as a 2D matrix such as [number of time chunks, number of features], and then use a set of 2D convolutional filters to extract high-level time-frequency features (which we refer to as SimpleNet-CNN).

In summary, SimpleNet is designed to avoid the aliasing effect problem and reduce unnecessary convolutional layers. It features a very concise architecture and high model interpretability. On the other hand, the proposed initializer allows us to add initial task-specific assumptions to the network.

## 4 EXPERIMENTS

### 4.1 SETUP

We perform our experiments on a 4-class speaker emotion recognition task (happy, sad, neutral, and angry) and a speaker gender recognition task. The speaker emotion recognition is challenging due to the complexity of the speech emotion patterns. By comparing this with speaker gender recognition (which is a simpler task), we can observe the behavior of the model to tasks of different complexity. Another reason we choose a speaker emotion recognition task is that WaveRNN is originally designed for this task, and hence it is fair to use this as a baseline for our experiments.

For our experiments, we use the audio part of the IEMOCAP dataset (Busso et al., 2008), which is a commonly used database in speech emotion recognition research. The IEMOCAP dataset is divided into five sessions, each session consisting of the conversations between a male and a female speaker. Speakers of different sessions are independent. We conduct leave-one-session-out 5-fold cross validation in all our experiments. A small validation set of 32 utterances is separated out from the testing set, and the best model is selected according to its performance on the validation set. The average length of utterances is 4.46s and we pad or cut all utterances into 6 seconds segments. The sampling rate of IEMOCAP is 16kHz, thus the waveform of each utterance is a 96000-dimensional vector.

We compare SoundNet, WaveRNN, SimpleNet-RNN, and SimpleNet-CNN in our experiments. SimpleNet-RNN and SimpleNet-CNN have identical front-end layers (number of filters: 64, length of filter: 256, mean absolute pooling). Their convolutional filters are initialized as a series of non-overlapping bandpass filters with the center frequency evenly distributed from 0Hz to 8000Hz. The high-level layers of SimpleNet-CNN are four identical 2-D convolutional layers with filter size (2,2) and filter number 32; each convolutional layer is followed by a max pooling layer with pooling size (2,2). The high-level layers of WaveRNN and SimpleNet-RNN are identical (two LSTM layers with 64 units). All fully-connected layers are identical with 64 units. All convolutional filters of SoundNet and WaveRNN are initialized with the Xavier initializer (Glorot & Bengio, 2010). All

networks except SoundNet have a windowing step with the window size of 40ms (thus there are 150 time chunks for each utterance). The LSTM units use the tanh activation, all other units are ReLU. Adam (Kingma & Ba, 2014) optimization is used in all experiment and the learning rate is searched in [1e-5, 1e-4, 1e-3].

In addition, we also include SpecNet, a spectrogram-based approach, which has high-level layers that are identical with SimpleNet-CNN and the input magnitude spectrogram is extracted in the preprocessing step using filters identical to the initialized filters of SimpleNet. That is, the only difference between SpecNet and SimpleNet-CNN is that SimpleNet-CNN has trainable convolutional filters. The comparison of the architecture of these networks is shown in Figure 5. Note that we intend to control certain parts of related networks identically (e.g., WaveRNN and SimpleNet-RNN have the same high-level layers; SimpleNet-CNN and SimpleNet-RNN have the same front-end layers), so that we can more clearly analyze the reason for performance differences.

## 4.2 RESULTS

Table 1: The accuracy (%) of the experiments (training accuracy is in parentheses)

|  | SoundNet | WaveRNN | SimpleNet-RNN | SimpleNet-CNN | SpecNet |
|---|---|---|---|---|---|
| Emotion Test | 48.7 (95.0) | 48.0 (83.8) | 49.2 (73.8) | 52.9 (80.0) | 42.3 (66.3) |
| Gender Test | 88.6 (99.4) | 88.8 (98.8) | 88.7 (96.3) | 88.6 (98.1) | 63.9 (78.1) |

Table 1 shows the results of our experiments, where we can observe that all waveform-based approaches perform similarly in the speaker gender recognition test, but demonstrate differences in the more complex speaker emotion recognition test.[1] For the emotion recognition test, the SimpleNet-CNN performs best, followed by SimpleNet-RNN. When comparing the models pair-wise, we obtain the following insights:

1) SimpleNet-RNN and WaveRNN have identical high-level layers, but SimpleNet-RNN performs better than WaveRNN, which shows that the concise front-end of SimpleNet is actually more effective than the 16-layer front-end of WaveRNN. This empirically proves our finding described in Section 2.2 that the stacked convolutional layers, especially those after the pooling layers, are unnecessary and ineffective.

2) SpecNet performs significantly worse than other waveform-based approaches. Since SpecNet has the same high-level layers with SimpleNet-CNN and its spectrogram is extracted using filters that are identical to the initialized filters of SimpleNet, the only reason for the performance difference is that SimpleNet-CNN has trainable convolutional filters, which can learn which spectra are informative during the training phase, while the spectrogram is extracted using fixed filters in SpecNet. Hence, the result shows that the contributions of trainable convolutional filters in waveform-based approaches are significant.

3) SimpleNet-CNN and SimpleNet-RNN have the same front-end layers, but SimpleNet-CNN performs better than SimpleNet-RNN. This shows that the high-level layers of SimpleNet-CNN are more effective than the LSTMs.

4) All networks have a certain amount of overfitting, but SoundNet and WaveRNN show more severe overfitting than SimpleNet-RNN and SimpleNet-CNN. This also empirically proves our discussion in Section 2.2 and Section 3 that the front-end layers of SoundNet and WaveRNN are less effective, leading the models to rely heavily on the dense layers, which further causes severe overfitting.

## 4.3 THE CONVOLUTIONAL FILTERS

As discussed in Section 2.2.2 and Section 3, the convolutional filters play an important role in waveform-based audio processing. It is very interesting to see how these filters are trained so that we can achieve a better understanding of the inner workings of CNNs. As described in Section 3

---

[1]In the gender recognition test, although the accuracy does not appear good enough for such a simple task, the low accuracy is actually caused by significant speaker variances among sessions and labeling errors (i.e., two speakers speak simultaneously during one utterance).

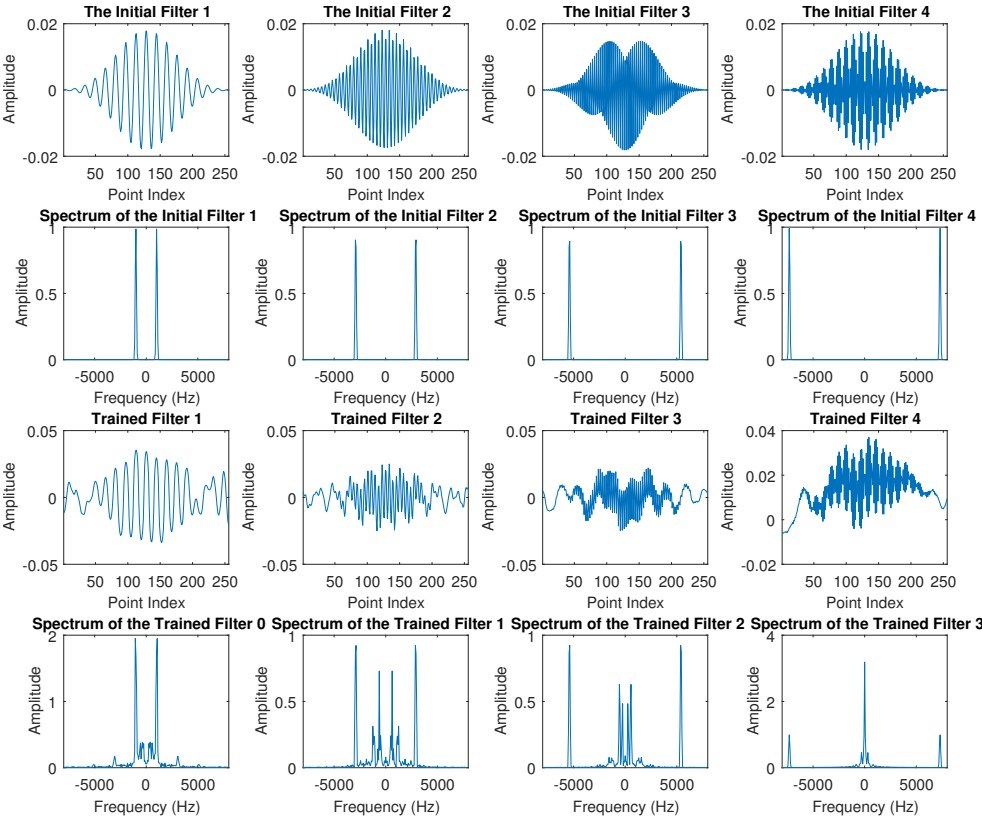

Figure 6: Visualization of 4 selected convolutional filters of SimpleNet-CNN before and after training in the speaker emotion recognition test.

and Section 4, the convolutional filters are initialized as a series of non-overlapping bandpass filters with center frequencies evenly distributed from 0Hz to 8000Hz before the training starts. This is an initial assumption provided to the CNNs, which means that each frequency band is of the same importance. However, we actually know that lower frequency components are more informative for speech, thus we are curious to see if the filters also discover this after training.

In Figure 6, we visualize 4 selected convolutional filters of SimpleNet-CNN in the time and frequency domains. The center frequency of these 4 selected filters ranges from low to high. We observe that the filter with different center frequencies has different learning patterns: the filters with high center frequencies (filters 2, 3, 4) change significantly from their initial states compared to the filter with low center frequency (filter 1) during training, and their change in the frequency domain is adding more passbands at low frequency. That is, the convolutional filters do actually find that lower frequency components are more informative, which is exactly what we expected.

## 5 CONCLUSIONS

A lack of understanding of how deep CNNs learn from audio waveforms hinders the development and further improvement of end-to-end audio processing technologies. In this work, we theoretically analyze and explain how deep CNNs learn from raw audio waveforms. Based on our analysis, we find that stacking convolutional layers are unnecessary and pooling layers can lead to the aliasing effect in audio processing, which are the potentially most significant contributors to the limited performance of existing solutions. Therefore, we propose a new network called SimpleNet, which features a very concise architecture and high model interpretability. Our experiments empirically prove our analysis and demonstrate the effectiveness of SimpleNet.

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
