# OpenReview forum: "How do deep convolutional neural networks learn from raw audio waveforms?"
_ICLR.cc/2018/Conference — Reject_

### Official Review · AnonReviewer3 · 2017-11-27
**Review of "How do deep convolutional neural networks learn from raw audio waveforms?"**

**Rating:** 3
**Confidence:** 4

**Review:**

Summary:

The authors aim to analyze what deep CNNs learn, and end up proposing “SimpleNet”, which essentially reduces the early feature extraction stage of the network to a single convolutional layer, which is initialized using pre-defined filters. The authors step through a specific example involving bandpass filters to illustrate that multiple layers of filtering can be reduced to a single layer in the linear case, as well as the limitations of pooling. Results on a 4-class speaker emotion recognition task demonstrate some advantage over other, deeper architectures that have been proposed, as well as  predefined feature processing.

Review:

The authors’ arguments and analysis are in my assessment rudimentary---the effects of pooling and cascading multiple linear convolutions are well appreciated by researchers in the field. Furthermore, the adaptation of “front-end” signal processing modules in and end-to-end manner has been considered extensively before (e.g. Sainath et al., 2015), and recent work on very deep networks for signal processing that shows gains on more substantial tasks have not been cited (e.g. Dai, Wei, et al. below). Finally, the experimental results, considering the extensive previous work in this area, are insufficient to establish novel performance in lieu of novel ideas.

Overall Recommendation:

Overall, this paper is a technical report that falls well below the acceptance threshold for ICLR for the reasons cited above. Reject.

---

> ### Author Response · Authors · 2017-12-16
> **The Authors' Respond**
>
> We thank the reviewer for the very helpful review of our paper! There are some points we would like to discuss with the reviewer (listed below). We would appreciate if the reviewer could provide us with another round of comments/suggestions.
>
> 1. The reviewer says our work is rudimentary since the effect of pooling/convolutional layers are well appreciated by researchers in the field. We do not agree with this statement; experimental improvements are of course important and meaningful, but when it conflicts with theoretical analysis, there must be something worth to study and investigate. Especially for widely used tools, we actually need more studies instead of completely trusting them.
>
> For example, in [1] (the reference provided), the 18-layer network actually has minor improvements compared to [2], which is a 4-layer network. The author of [1] states that the limited improvement is due to 1) [2] uses 10-folder cross-validation, [2] uses hold-out validation. There should not be much of a difference and 10-folder cross-validation is actually more reasonable. Thus, we do not think that this is a strong reason. 2) [2] uses 44100Hz recordings, [1] uses 8000Hz recording. If this is the reason limiting the achievable improvement, this means that downsampling will lower the performance, which proves our point to ‘not using pooling layers aggressively’.
>
> Another example is in Table.6 of [3]; the authors also report that the output of the middle layer (rather than the last layer) of the network is the most discriminative feature, which shows more layers make the representation worse and thus also partly proves our conclusion that ‘stacking convolutional/pooling layers is not always effective’.
>
> In summary, current studies on using deep learning to process raw waveforms achieve performance improvement empirically, but we also see some phenomena that cannot be explained. In this work, we try to analyze it from the signal processing perspective, which is proved by our experiment, and our analysis can also explain some phenomena of previous work. Therefore, we do not think that this work is rudimentary.
>
> 2. Reference/experiment. We apologize for missing some references in this paper. However, due to the very large volume of works that use DNNs to process raw waveforms, we are not able to cite all them in our paper. Similarly,  there are a large number of tasks in the audio field, including speech recognition, speech emotion recognition, speaker recognition, and environment recognition. We are not able to show experiments for all these tasks. Therefore, we would appreciate suggestions from the reviewer what kind of experiments could make our conclusions most convincing to allow us to improve this work.
>
> [1] Dai, W., Dai, C., Qu, S., Li, J., & Das, S. (2016). Very deep convolutional neural networks for raw waveforms. arXiv preprint arXiv:1610.00087.
>
> [2] Piczak, K. J. (2015, September). Environmental sound classification with convolutional neural networks. In Machine Learning for Signal Processing (MLSP), 2015 IEEE 25th International Workshop on (pp. 1-6). IEEE.
>
> [3] Aytar, Y., Vondrick, C., & Torralba, A. (2016). Soundnet: Learning sound representations from unlabeled video. In Advances in Neural Information Processing Systems (pp. 892-900).

---

### Official Review · AnonReviewer1 · 2017-11-27

**Rating:** 3
**Confidence:** 5

**Review:**

The paper proposes a CNN-based based approach for speech processing using raw waveforms as input. An analysis of convolution and pooling layers applied on waveforms is first presented. An architecture called SimpleNet is then presented and evaluated on two speech tasks: emotion recognition and gender classification.

This paper propose a theoretical analysis of convolution and pooling layers to motivate the SimpleNet architecture. To my understanding, the analysis is flawed (see comments below). The SimpleNet approach is interesting but not sufficiently backed with experimental results. The network analysis is minimal and provides almost no insights. I therefore recommend to reject the paper.

Detailed comments:

Section 1:

* “Therefore, it remains unknown what actual features CNNs learn from waveform”. This is not true, several works on speech recognition have shown that a convolution layer taking raw speech as input can be seen as a bank of learned filters. For instance in the context of speech recognition, [9] showed that the filters learn phoneme-specific responses, [10] showed that the learned filters are close to Mel filter banks and [7] showed that the learned filters are related to MRASTA features and Gabor filters. The authors should discuss these previous works in the paper.

Section 2:

* Section 2.1 seems unnecessary, I think it’s safe to assume that the Shannon-Nyquist theorem and the definition of convolution are known by the reader.

* Section 2.2.2 & 2.2.3: I don't follow the justification that stacking convolutions are not needed: the example provided is correct if two convolutions are directly stacked without non-linearity, but the conclusion does not hold with a non-linearity and/or a pooling layer between the convolutions: two stacked convolutions with non-linearities are not equivalent to a single convolution. To my understanding, the same problem is present for the pooling layer: the presented conclusion that pooling introduces aliasing is only valid for two directly stacked pooling layers and is not correct for stacked blocks of convolution/pooling/non-linearity.

* Section 2.2.5: The ReLU can be seen as a half-wave rectifier if it is applied directly to the waveform. However, it is usually not the case as it is applied on the output of the convolution and/or pooling layers. Therefore I don’t see the point of this section.

* Section 2.2.6: In this section, the authors discuss the differences between spectrogram-based and waveforms-based approaches, assuming that spectrogram-based approach have fixed filters. But spectrogram can also be used as input to CNNs (i.e. using learned filters) for instance in speech recognition [1] or emotion recognition [11]. Thus the comparison could be more interesting if it was between spectrogram-based and raw waveform-based approaches when the filters are learned in both cases.

Section 3:

* Figure 4 is very interesting, and is in my opinion a stronger motivation for SimpleNet that the analysis presented in Section 2.

* Using known filterbanks such as Mel or Gammatone filters as initialization point for the convolution layer is not novel and has been already investigated in [7,8,10] in the context of speech recognition.

Section 4:

* On emotion recognition, the results show that the proposed approach is slightly better, but there is some issues: the average recall metric is usually used for this task due to class imbalance (see [1] for instance). Could the authors provide results with this metric ? Also IEMOCAP is a well-used corpus for this task, could the authors provide some baselines performance for comparison (e.g. [11]) ?

* For gender classification, there is no gain from SimpleNet compared to the baselines. The authors also mention that some utterances have overlapping speech. These utterances are easy to find from the annotations provided with the corpus, so it should be easy to remove them for the train and test set. Overall, in the current form, the results are not convincing.

* Section 4.3: The analysis is minimal: it shows that filters changed after training (as already presented in Figure 4). I don't follow completely the argument that the filters should focus on low frequency. It is more informative, but one could expect that the filters will specialized, thus some of them will focus on high frequencies, to model the high frequency events such as consonants or unvoiced event.
It could be very interesting to relate the learned filters to the labels: are some filters learned to model specific emotions ? For gender classification, are some filters focusing on the average pitch frequency of male and female speaker ?

* Finally, it would be nice to see if the claims in Section 2 about the fact that only one convolution layer is needed and that stacking pooling layers can hurt the performance are verified experimentally: for instance, experiments with more than one pair of convolution/pooling could be presented.

Minor comments:

* More references for raw waveforms-based approach for speech recognition should be added [3,4,6,7,8,9] in the introduction.

* I don’t understand the first sentence of the paper: “In the field of speech and audio processing, due to the lack of tools to directly process high dimensional data …”. Is this also true for any pattern recognition fields ?

* For the MFCCs reference in 2.2.2, the authors should cite [12].

* Figure 6: Only half of the spectrum should be presented.

References:

[1] H. Lee, P. Pham, Y. Largman, and A. Y. Ng. Unsupervised feature learning for audio classification using convolutional deep belief networks. In Advances in Neural Information Processing Systems 22, pages 1096–1104, 2009.

[2] Schuller, Björn, Stefan Steidl, and Anton Batliner. "The interspeech 2009 emotion challenge." Tenth Annual Conference of the International Speech Communication Association. 2009.

[3] N. Jaitly, G. Hinton, Learning a better representation of speech sound waves using restricted Boltzmann machines, in: Proceedings of the IEEE International Conference on Acoustics, Speech and Signal Processing (ICASSP), 2011, pp. 5884–5887.

[4] D. Palaz, R. Collobert, and M. Magimai.-Doss. Estimating Phoneme Class Conditional Probabilities from Raw Speech Signal using Convolutional Neural Networks, INTERSPEECH 2013, pages 1766–1770.

[5] Van den Oord, Aaron, Sander Dieleman, and Benjamin Schrauwen. "Deep content-based music recommendation." Advances in neural information processing systems. 2013.

[6] Z.Tuske, P.Golik, R.Schluter, H.Ney, Acoustic Modeling with Deep Neural Networks Using Raw Time Signal for LVCSR,
in: Proceedings of the Annual Conference of the International Speech Communication Association (INTERSPEECH), Singapore, 2014, pp. 890–894.

[7] P. Golik, Z. Tuske, R. Schlu ̈ter, H. Ney, Convolutional Neural Networks for Acoustic Modeling of Raw Time Signal in LVCSR, in: Proceedings of the Annual Conference of the International Speech Communication Association (INTERSPEECH), 2015, pp. 26–30.

[8] Yedid Hoshen and Ron Weiss and Kevin W Wilson, Speech Acoustic Modeling from Raw Multichannel Waveforms, International Conference on Acoustics, Speech, and Signal Processing, 2015.

[9] D. Palaz, M. Magimai-Doss, and R. Collobert. Analysis of CNN-based Speech Recognition System using Raw Speech as Input, INTERSPEECH 2015, pages 11–15.

[10] T. N. Sainath, R. J. Weiss, A. Senior, K. W. Wilson, and O. Vinyals. Learning the Speech Front-end With Raw Waveform CLDNNs. Proceedings of the Annual Conference of the International Speech Communication Association (INTERSPEECH), 2015.

[11] Satt, Aharon & Rozenberg, Shai & Hoory, Ron. (2017). Efficient Emotion Recognition from Speech Using Deep Learning on Spectrograms. 1089-1093. Interspeech 2017.

[12] S. Davis and P. Mermelstein. Comparison of parametric representations for monosyllabic word recognition in continuously spoken sentences. IEEE Transactions on Acoustics, Speech and Signal Processing, 28(4):357–366, 1980.

---

> ### Author Response · Authors · 2017-12-16
> **The Authors' Respond (Part 2)**
>
> 5. About the dataset and experiment, similar to other works, we combine happy and excited together as the new happy class, which makes the classes roughly balanced. The IEMOCAP database is not designed for gender recognition, thus some utterances contain multiple speakers, but are only annotated as a single speaker. We would greatly appreciate any suggestions by the reviewer on how to avoid this issue. In our experiments, we conducted a strict leave-one-session-out evaluation strategy. In [8], the authors use the same evaluation strategy and achieved 52.84%, which is similar to SimpleNet (52.9%), while SimpleNet is a simpler approach.
>
> [1] Golik, P., Tüske, Z., Schlüter, R., & Ney, H. (2015). Convolutional neural networks for acoustic modeling of raw time signal in LVCSR. In Sixteenth Annual Conference of the International Speech Communication Association.
>
> [2] Hoshen, Y., Weiss, R. J., & Wilson, K. W. (2015, April). Speech acoustic modeling from raw multichannel waveforms. In Acoustics, Speech and Signal Processing (ICASSP), 2015 IEEE International Conference on (pp. 4624-4628). IEEE.
>
> [3] Sainath, T. N., Weiss, R. J., Senior, A., Wilson, K. W., & Vinyals, O. (2015). Learning the speech front-end with raw waveform CLDNNs. In Sixteenth Annual Conference of the International Speech Communication Association.
>
> [4] Aytar, Y., Vondrick, C., & Torralba, A. (2016). Soundnet: Learning sound representations from unlabeled video. In Advances in Neural Information Processing Systems (pp. 892-900).
>
> [5] Zeiler, M. D., & Fergus, R. (2014, September). Visualizing and understanding convolutional networks. In European conference on computer vision (pp. 818-833). Springer, Cham.
>
> [6] Lee, H., Pham, P., Largman, Y., & Ng, A. Y. (2009). Unsupervised feature learning for audio classification using convolutional deep belief networks. In Advances in neural information processing systems (pp. 1096-1104).
>
> [7] Satt, A., Rozenberg, S., & Hoory, R. (2017). Efficient Emotion Recognition from Speech Using Deep Learning on Spectrograms. Proc. Interspeech 2017, 1089-1093.
>
> [8] Ghosh, S., Laksana, E., Morency, L. P., & Scherer, S. (2016). Representation Learning for Speech Emotion Recognition. In INTERSPEECH (pp. 3603-3607).

---

> ### Author Response · Authors · 2017-12-16
> **The Authors' Respond (Part 1)**
>
> We thank the reviewer for the very thorough and helpful review (especially the extensive references)! There are some points we would like to discuss with the reviewer (listed below). We would appreciate if the reviewer can give us another round of comments/suggestions.
>
> 1. In [1,2,3], the analysis was focused on only the first and second convolution layers (since these networks are relatively shallow). For deeper networks, there is no investigation yet of the functioning and inner workings of deeper layers, e.g., in [4], the authors only analyze the first convolutional layer, while SoundNet has 8 layers. Further, in Table.6 of [4], the authors also report that the output of the middle layer (rather than the last layer) is the most discriminative feature, but they do not provide an analysis. In fact, it shows that more layers make the representation worse and thus partly proves our conclusion. In computer vision research, the functioning of the first convolutional layer is trivial, but the functioning of deeper layers was not clear until the work in [5] was presented. Thus, we believe that the study of the inner workings of deeper layers is different from previous research, but also very interesting.
>
> 2. The reviewer argues that ReLU can be seen as a half-wave rectifier only if it is applied directly to the waveform, not the output of the convolutional/pooling layer. In fact, there is no real ‘original’ waveform. The waveform we input to the DNNs has already been explicitly or implicitly filtered/downsampled by the recording device. The convolutional/pooling layer just performs another round of filtering/downsampling. Thus, the output of the convolutional/pooling layer is still a temporal signal and follows the basic rules of signal processing. Hence, ReLU still can be regarded as the half-wave rectifier when it is applied to the output of mid-layers. Another concern of the reviewer is that aliasing is not valid for the output of stacked blocks of convolutional/pooling/non-linearity; for the same reason stated above, the output of the stacked blocks is still a temporal signal, which will be affected by the aliasing effect.
>
> Note that this analysis is from the perspective of signal processing. We actually can think of the process of front-end layers from two different perspectives: the perspective of machine learning and the perspective of signal processing. From the perspective of machine learning, the deeper architecture and the non-linearities all add to the expressivity (capacity) of the model and thus possibly (but not always) help improve the performance. From the perspective of signal processing, the processing of the front-end layers should extract useful information from the signal, but at least not destroy the information of the signal. The aggressive pooling layer/ aliasing effect actually destroys the information of the signal (this is similar to aggressively downsampling the input waveform), which is a strong motivation for SimpleNet.
>
> 3. The review is correct that stacking convolutional layers with non-linearity cannot be replaced by a single convolutional layer. But as stated in Section 2.2.5, the most widely used non-linearity ReLU, from the perspective of signal processing, only adds harmonic frequency components to the signal, which leads to a distortion (rather than a meaningful feature extraction). Thus, we do have reason to doubt the effectiveness of stacking convolutional layers.
>
> 4. The reviewer says the spectrogram can be used as input for CNNs [6,7]. This is correct, and we do compare to specNet in our experiment. The architecture of specNet is very similar to the architecture proposed in [7]. Note that due to the limited number of FFT points, the spectrogram usually have fixed and a limited number of FFT bins; suppose the spectrogram has 80 FFT bins for [0Hz,8000Hz] and the average energy/magnitude of each 100Hz range will be placed in the same bin, then the difference of each spectra components within this 100Hz range cannot be recognized in the further processing. Also, the size/number of bins are fixed. In contrast, for the raw waveform approach, the learnable filters (with enough number of points) can perform flexible filtering.

---

### Official Review · AnonReviewer2 · 2017-11-28
**useful idea, but experimental validation and analysis need to be much stronger**

**Rating:** 2
**Confidence:** 5

**Review:**

The paper provides an analysis of the representations learnt in convolutional neural networks that take raw audio waveforms as input for a speaker emotion recognition task. Based on this analysis, an architecture is proposed and compared to other architectures inspired by other recent work. The proposed architecture overfits less on this task and thus performs better.

I think this work is not experimentally strong enough to draw the conclusions that it draws. The proposed architecture, aptly called "SimpleNet", is relatively shallow compared to the reference architectures, and the task that is chosen for the experiments is relatively small-scale. I think it isn't reasonable to draw conclusions about what convnets learn in general from training on a single task, and especially not a small-scale one like this.

Moreover, SoundNet, which the proposed architecture is compared to, was trained on (and designed for) a much richer and more challenging task originally. So it is not surprising at all that it overfits dramatically to the tasks chosen here (as indicated in table 1), and that a much shallower network with fewer parameters overfits less. This seems obvious to me, and contrary to what's claimed in the paper, it provides no convincing evidence that shallow architectures are inherently better suited for raw audio waveform processing. This is akin to saying that LeNet-5 is a better architecture for image classification than Inception, because the latter overfits more on MNIST. Perhaps using the original SoundNet task, which is much more versatile, would have lent some more credibility to these claims.

The analysis in section 2.2 is in-depth, but also not very relevant: it ignores the effects of nonlinearities, which are an essential component of modern neural network architectures. Studying their effects in the frequency domain would actually be quite interesting. It is mentioned that the ReLU nonlinearity acts as a half-wave rectifier, but the claim that its effect in the frequency domain is small compared to aliasing is not demonstrated. The claim that "ReLU and non-linear activations can improve the network performance, but they are not the main factors in the inner workings of CNNs" is also unfounded.

The conclusion that stacking layers is not useful might make sense in the absence of nonlinearities, but when each layer includes a nonlinearity, the obvious point of stacking layers is to improve the expressivity of the network. Studying aliasing effects in raw audio neural nets is a great idea, but I feel that this work takes some shortcuts that make the analysis less meaningful.




Other comments:

The paper is quite lengthy (11 pages of text) and contains some sections that could easily be removed, e.g. 2.1.1 through 2.1.3 which explain basic signal processing concepts and could be replaced by a reference. In general, the writing could be much more concise in many places.

The paper states that "it remains unknown what actual features CNNs learn from waveforms.". There is actually some prior work that includes some analysis on what is learnt in the earlier layers of convnets trained on raw audio:
"Learning the Speech Front-end With Raw Waveform CLDNNs", Sainath et al.
"Speech acoustic modeling from raw multichannel waveforms", Hoshen et al.
"End-to-end learning for music audio", Dieleman & Schrauwen
Only the first one is cited, but not in this context. I think saying "it remains unknown" is a bit too strong of an expression.

The meaning of the following comment is not clear to me: "because in computer vision, the spatial frequency is not the only information the model can use". Surely the frequency domain and the spatial domain are two different representations of the same information contained in an image or audio signal? So in that sense, spatial frequency does encompass all information in an image.

The implication that high-frequency information is less useful for image-based tasks ("the spatial frequency of images is usually low") is incorrect. While lower frequencies dominate the spectrum more obviously in images than in audio, lots of salient information (i.e. edges, textures) will be high-frequency, so models would still have to learn high-frequency features to perform useful tasks.

WaveNet is mentioned (2.2.4) but not cited. WaveNet is a fairly different architecture than the ones discussed in this paper and it would be useful to at least discuss it in the related work section. A lot of the supposed issues discussed in this paper don't apply to WaveNet (e.g. there are no pooling layers, there is a multiplicative nonlinearity in each residual block).

The paper sometimes uses concepts without clearly defining them, e.g. "front-end layers". Please clearly define each concept when it is first introduced.

The paper seems to make a fairly arbitrary distinction between layers that perform signal filtering operations, and layers that don't - but every layer can be seen as a (possibly nonlinear) filtering operation. Even if SimpleNet has fewer "front-end layers", surely the later layers in the network can still introduce aliasing? I think the implicit assumption that later layers in the network perform a fundamentally different kind of operation is incorrect.

It has been shown that even random linear filters can be quite frequency-selective (see e.g. "On Random Weights and Unsupervised Feature Learning", Saxe et al.). This is why I think the proposed "changing rate" measure is a poor choice to show effective training. Moreover, optimization pathways don't have to be linear in parameter space, and oscillations can occur. Why not measure the difference with the initial values (at iteration 0)? It seems like that would prove the point a bit better.

Manually designing filters to initialize the weights of a convnet has been done in e.g. Sainath et al. (same paper as mentioned before), so it would be useful to refer to it again when this idea is discussed.

In SpecNet, have the magnitude spectrograms been log-scaled? This is common practice and it can make a dramatic difference in performance. If you haven't tried this, please do.

---

> ### Author Response · Authors · 2017-12-16
> **The Authors' Respond (Part 2)**
>
> 4. The reviewer is correct that high-frequency information is important in image-based tasks, and we are not denying this point. For images, the spatial frequency is relatively low compared to the aliasing threshold, and thus, the aliasing effect is less likely to happen when the images are down-sampled. Images with high spatial frequency can be found at https://web.stanford.edu/class/ee368b/Projects/panu/pages/aliasing.html; natural images, e.g., images in ImageNet, rarely have such high spatial frequency. In contrast,  audio waveforms usually cannot handle aggressive down-sampling, because the sampling rate is usually chosen as the minimum value to keep the useful information and avoid aliasing, e.g., for human speech that was sampled at 16,000Hz, the sound quality will substantially drop after a pooling layer with pooling size of 4. The reason to do this comparison is to argue that we should not always follow the approach used to design DNNs for images when designing DNNs for audios, because they are different representations.
>
> References:
>
> [1] Trigeorgis G, Ringeval F, Brueckner R, et al. Adieu features? End-to-end speech emotion recognition using a deep convolutional recurrent network[C]//Acoustics, Speech and Signal Processing (ICASSP), 2016 IEEE International Conference on. IEEE, 2016: 5200-5204.
>
> [2] Schuller, B., Steidl, S., Batliner, A., Bergelson, E., Krajewski, J., Janott, C., ... & Warlaumont, A. S. (2017, February). The INTERSPEECH 2017 computational paralinguistics challenge: Addressee, cold & snoring. In Computational Paralinguistics Challenge (ComParE), Interspeech 2017.
>
> [3] Aytar, Y., Vondrick, C., & Torralba, A. (2016). Soundnet: Learning sound representations from unlabeled video. In Advances in Neural Information Processing Systems (pp. 892-900).

---

> > ### Comment · AnonReviewer2 · 2018-01-12
> > **rebuttal response**
> >
> > 0. Noted, thanks for clarifying this.
> >
> > 1. I'm still not convinced. You say that "the processing of raw waveforms will also go through a similar time-feature representation", but how do you know where that is? At which point in the network do you call something a "time-feature representation"? Isn't every layer in the network technically producing a time-feature representation? I've actually come to suspect you're talking about the point in the network where the time dimension completely disappears (e.g. through pooling), but you should definitely state this more clearly in that case. It is incorrect and confusing to call one type of representation a "temporal signal" and another "features". A temporal signal also consists of (a sequence of) features.
> >
> > 2. I think Section 2.2.3 does not show convincingly that useful information is destroyed at all. I don't dispute that there will be aliasing effects, but these are only problematic when looking at each feature in isolation. A neural network layer produces a vector of complementary features, and these can disambiguate each other even if a lot of aliasing occurs in each feature individually. So I still disagree fundamentally with this argument.
> >
> > 3. To compare models fairly, first choose a basis of comparison, e.g. model capacity (which could be expressed in # learnable parameters) or inference speed, or something else. I think the former is most appropriate here. Once you fix the number of parameters it should be pretty straightforward to compare a few different architectures of different depths.
> >
> > 4. While I agree that the spectral properties of audio signals and images are very different, most salient information in images is high-frequency, and the point of most neural nets is precisely to pick up on this salient information. Since pooling layers work just fine in discriminative networks trained on images (admittedly they are not strictly necessary, either), I don't think this argument makes sense either. Sure, pooling filtered audio signals will remove information, but you simply cannot ignore the interaction with nonlinearities here, nor the fact that that you always have multiple feature signals that are complementary to each other and can disambiguate each other.
> >
> >
> > I've decided not to change my rating as I think almost all of my criticisms still hold for the current version of the paper.

---

> ### Author Response · Authors · 2017-12-16
> **The Authors' Respond (Part 1)**
>
> We thank the reviewer for the very thorough and helpful review of the paper! There are some points we would like to discuss with the reviewer (listed below) and we would appreciate it if the reviewer could give us another round of comments/suggestions.
>
> 0. The ‘waveNet’ in Section 2.2.4 is a typo, we actually meant the network proposed in [1], which we refer as waveRNN in the paper, this network structure is also used as a baseline in [2] for a variety of tasks, thus we chose to compare our work to waveRNN.
>
> 1. The definition of ‘front-end layers’. A spectrogram can be regarded as a time-frequency representation, i.e., a matrix of shape [num_time_steps, num_frequency_bins], each element of the matrix is the magnitude or energy of the frequency component. If we regard each element as a feature, we can also consider it as a time-feature representation. As described in Section 2.2.1, no matter if we explicitly use windowing or implicitly fewer pooling layers, the processing of raw waveforms will also go through a similar time-feature representation, as shown in the white box in Figure 5. We define the layers before the time-feature representation as front-end layers and the layers after the time-feature representation as high-level layers. One of the reviewer’s concern is that the high-level layers will also introduce aliasing effects. However, this is not the case, because the aliasing effect only impacts the temporal signal; the high-level layers are processing the features.
>
> 2.The purpose of the work is not to claim that shallow networks can outperform deep networks; actually, in our experiments, when we compare SimpleNet-CNN and SimpleNet-RNN, we show that deeper high-level architectures can be beneficial. Instead, we doubt the effectiveness of deep front-end architectures. We actually can think of the process of front-end layers from two different perspectives: the perspective of machine learning and the perspective of signal processing. From the perspective of machine learning, the deeper architecture and the non-linearities all add to the expressivity (capacity) of the model and thus possibly (but not always) help improve the performance. From the perspective of signal processing, the processing of front-end layers should extract useful information from the signal, but at least not destroy the information in the signal. As shown in Section 2.2.3, the pooling layer/aliasing effect destroys the useful information unrecoverably. ReLU adds harmonic frequency components, which also leads to distortion. Thus, from the perspective of signal processing, it is difficult to support the claim that deep front-end structures are useful.
>
> 3. We agree with the reviewer that more experiments (e.g., more tasks and more datasets) will make our conclusion more convincing. However, we do not agree that the overfitting problem makes the comparison unfair. In fact, one can always argue that the dataset is too small for large networks. Then the comparison of deep and shallow networks will be impossible. The IEMOCAP dataset we use has more than 5000 utterances in the selected 4 classes, which is at least large enough for waveRNN, because waveRNN was originally tested using a similar size dataset. SoundNet uses a much larger training set since it is semi-unsupervised, but even using such a large training set, in Table.6 of [3], the authors also report that the output of the middle layer (rather than the last layer) is the most discriminative feature, which shows more layers make the representation worse and therefore partly supports our conclusion. In addition, we use early stop for all experiments. If the reviewer could provide additional information on how to make the comparison more fair for networks with different numbers of layers (and what kinds of experiments would make our conclusion more convincing), this would greatly help us with improving our work.

---

### Author Response · Authors · 2018-01-04
**Paper revision**

We fixed the typo in the Section 2.2.4, and adjusted the references according to the reviewers' suggestion in the new version.

---

### Decision · Program_Chairs · 2018-01-29
**ICLR 2018 Conference Acceptance Decision**

**Decision:**

Reject

**Comment:**

The reviewers rightly point out that presented analysis is limiting and that the experimental results are not extensive enough. Moreover, several existing work that use raw waveforms have interesting analysis of what the network is trying to learn. Given these comments, the AC recommends that the paper be rejected.